# Machine learning identifies risk factors associated with long-term sick leave following COVID-19 in Danish population

Kim Daniel Jakobsen [1] ✉, Elisabeth O'Regan [1], Ingrid Bech Svalgaard[1] & Anders Hviid [1,2]

## Abstract

**Background** Post COVID-19 condition (PCC) can lead to considerable morbidity, including prolonged sick-leave. Identifying risk groups is important for informing interventions. We investigated heterogeneity in the effect of SARS-CoV-2 infection on long-term sick-leave and identified subgroups at higher risk.

**Methods** We conducted a hybrid survey and register-based retrospective cohort study of Danish residents who tested positive for SARS-CoV-2 between November 2020 and February 2021 and a control group who tested negative, with no known history of SARS-CoV-2. We estimated the causal risk difference (RD) of long-term sick-leave due to PCC and used the causal forest method to identify individual-level heterogeneity in the effect of infection on sick-leave. Sick-leave was defined as >4 weeks of full-time sick-leave from 4 weeks to 9 months after the test.

**Results** Here, in a cohort of 88,818 individuals, including 37,482 with a confirmed SARS-CoV-2 infection, the RD of long-term sick-leave is 3.3% (95% CI 3.1% to 3.6%). We observe a high degree of effect heterogeneity, with conditional RDs ranging from −3.4% to 13.7%. Age, high BMI, depression, and sex are the most important variables explaining heterogeneity. Among three-way interactions considered, females with high BMI and depression and persons aged 36–45 years with high BMI and depression have an absolute increase in risk of long-term sick-leave above 10%.

**Conclusions** Our study supports significant individual-level heterogeneity in the effect of SARS-CoV-2 infection on long-term sick-leave, with age, sex, high BMI, and depression identified as key factors. Efforts to curb the PCC burden should consider multimorbidity and individual-level risk.

## Plain language summary

The burden of post COVID-19 condition varies from one person to another due to individual characteristics such as age, sex, and having single- or multiple pre-existing conditions. Sick leave following initial SARS-CoV-2 infection is one way to quantify this burden. However, to what extent the combinations of these characteristics impact the risk of post-acute sick leave is not well understood. Here, using a machine learning method, we observe that persons infected with SARS-CoV-2 have an increased risk of taking long-term sick leave compared to persons with no history of infection. Age, high BMI, sex, and depression explained substantial effect variation on the risk of long-term sick leave after infection. This knowledge may be used to help inform patient-targeted interventions.

[1] Department of Epidemiology Research, Statens Serum Institut, Copenhagen, Denmark. [2] Pharmacovigilance Research Centre, Department of Drug Design and Pharmacology, University of Copenhagen, Copenhagen, Denmark. ✉email: kija@ssi.dk

Most people infected with SARS-CoV-2 recover with no post-acute effects. However, an estimated 10–20% experience post-acute symptoms lasting at least 2 months, termed post COVID-19 condition (PCC) by the World Health Organization[1,2]. Thus, PCC represents a substantial health burden for both society and the individual. PCC is a complex heterogeneous disorder[3,4], with ongoing research into understanding risk groups and potential mechanisms[5–8].

Sick leave has been identified as a potential indicator of the burden of PCC[9–11] and represents an objective measure of societal impact and the impact on daily living for working age individuals. A significant increase in the burden of full-time post-acute sick leave was found in the general Danish population during the index wave in a period 1–9 months after infection with SARS-CoV-2[12], using the nationwide Danish survey, EFTER-COVID[13]. The study reports an increase in risk of taking a substantial amount of post-acute sick leave (defined as >1 month during the 8 months of post-acute follow-up) of 3.3% (95% CI: 3.0% to 3.5%) when infected with SARS-CoV-2. Large increases in risk were found across several single risk factors, e.g. fibromyalgia (RD 10.6%, 95% CI: 7.0% to 14.6%).

Due to the many unknowns of PCC, it is difficult to pre-specify subgroups for analyses when trying to identify those susceptible to developing PCC. Meta-analyses have shown that female sex, older age, obesity, smoking, and several health conditions are independently associated with an increased risk of developing PCC[14–16]. However, the use of a data-driven approach is necessary to identify novel subgroups and provide patient-centred estimates of risk that take all patient characteristics into account at the same time in contrast to conventional interaction analysis.

Patient-centred estimates of risk are particularly useful for public health strategy planning and policy making. As an example, several studies support that vaccination offers protection against PCC[14,17]. As of autumn/winter 2022-23, in Denmark, booster vaccination has only been recommended for people over 50 years and risk groups at higher risk of severe COVID-19 outcomes[18]. Moving forward with such recommendations, it may also be important to consider PCC risk, necessitating identification of those at higher risk of developing PCC.

In this study, we look for possible PCC risk groups using the EFTER-COVID survey by investigating how the effect of COVID-19 infection on the risk of taking substantial post-acute full-time sick leave depends on multiple characteristics and pre-existing health conditions. This is done using a state-of-the-art machine learning method for causal inference to estimate conditional risk differences (CRDs). Specifically, we use the causal forest (CF) method, a special case of the recently developed generalised random forest (GRF) machine learning method[19]. We find that persons infected with SARS-CoV-2 have an increased risk of taking substantial post-acute sick leave compared to persons with no history of infection. Age, high body mass index, sex, and depression show substantial effect variation on the risk of post-acute sick leave.

## Methods

### Data sources and study population
This was a retrospective cohort study, where we merged data from the EFTER-COVID survey with register data. The EFTER-COVID survey was designed to monitor PCC in the Danish population[13]. The present study includes participants who responded to a retrospective questionnaire sent out 9 months after being tested for SARS-CoV-2 using the web-based questionnaire service SurveyXact. An individual was eligible for invitation to the retrospective questionnaire if they were alive and living in Denmark 9 months after the test date, had a first positive RT-PCR test or a negative RT-

PCR test taken between November 4, 2020 and February 1, 2021, didn't have a positive test result <9 months after the test date, and was registered with the national mail system, e-Boks, used by 90% of Danish residents aged ≥15 years. Invitations were sent out to all eligible individuals with a positive test result, while test-negative controls were randomly selected using incidence density sampling on the test date with a ratio of 2:3 between test-positive and -negative persons. After receiving an invitation, participants were excluded if they failed to complete the questionnaire, indicated they believed they had been previously infected with SARS-CoV-2 due to receiving a seropositive result for SARS-CoV-2, or were >65 years-old at the time of the test.

The survey data was merged with Danish register data using the unique identifier (the CPR-number) in the Danish Civil Registration System assigned to all Danish residents.

The present study builds upon previous work by colleagues which explored the association between COVID-19 post-acute full-time sick leave and the possible impact of single risk factors[12]. Both of these studies examine the same cohort but with different analytical approaches.

### Exposure ascertainment
SARS-CoV-2 infection was ascertained using reverse transcription polymerase chain reaction (RT-PCR) tests, recorded in the Danish microbiology database (MiBa). RT-PCR tests were available and accessible for all adults free of charge and independent of the indication for acquiring a test[20]. Additionally, persons admitted to hospitals were tested for SARS-CoV-2.

### Outcome ascertainment
Participants were asked if they took sick leave around the time of their test date (indicated in the questionnaire) or any time since then. Participants could indicate sick leave up to 4 weeks, or >4 weeks, after the test. If sick leave was indicated >4 weeks after the test, further questions were asked about the type of sick leave (part-time or full-time), and the duration of each type of sick leave. This study considers substantial post-acute full-time sick leave, defined as taking >4 weeks of full-time sick leave in the period from 4 weeks after the test to 9 months after the test.

### Other covariates
Information on age and sex was obtained from the CPR-registry. Education and BMI were obtained from the questionnaire. Participants were asked about their highest completed education. The possible answers were Primary/elementary school (9th-10th grade), general secondary education or vocational education, vocational training, shorter term higher education (1–2 years), medium term higher education (2–4 years), longer term higher education (>5 years), and don't know/none of the above/do not wish to answer. We define high BMI as a body mass index (BMI) above 30 for persons aged 18 years or above, and for persons aged 15–17 years international cut-off points for obesity by age and sex were used[21]. BMI was calculated as response on weight in kilograms divided by response on height in meters squared. Response to questions on height and weight were not required, and in case of either missing, high BMI was reported as unknown. Information on health conditions were obtained from the questionnaire, in which participants were asked about existing health conditions diagnosed by a doctor before the RT-PCR test date. From the Danish National Patient Register (DNPR), information was obtained on in- and outpatient diagnoses coded using the 10th revision of the International Statistical Classification of Diseases and Related Problems (ICD-10), which enabled the calculation of the Charlson Comorbidity Index.

**Statistical analysis**. We used the causal forest (CF) method to estimate conditional risk differences (CRD). Specifically, we estimated the increase in risk of taking substantial full-time sick leave in the post-acute phase, conditioned on individual characteristics and pre-existing comorbidities, when infected with SARS-CoV-2, compared with not being infected. The causal forest was grown with splits allowed on the following covariates: age, sex, Charlson Comorbidity Index, education level, chronic asthma, diabetes, high blood pressure, COPD or other chronic lung disease, chronic or frequent headaches/migraines, fibromyalgia, chronic fatigue syndrome, anxiety, depression, post-traumatic stress disorder (PTSD), and high BMI. Age was included as a continuous covariate while the remaining covariates were included using one-hot encoding. The model used 2000 causal trees. Parts of a causal tree used by the model can be found in Supplementary Fig. 1. To reduce bias in tree predictions, we employed a subsample splitting technique called honesty. In this approach, the random subsample used for each tree is further split into two subsamples. One is used to construct the tree structure, and the other is used to populate the end nodes (leaves) of the tree. Empty leaves are pruned after repopulating with the second subsample. Default values from the *grf* R-package were used for most of the algorithm's tuning parameters, including the fraction of each tree sample assigned to the two honesty subsamples (0.5), and the number of covariates to try for each split ($\sqrt{p} + 20$). The maximum split imbalance was set to 0.01 to allow splits on rare health conditions, while the maximum node size was set to 10 to limit the computational requirements.

The CF CRD estimates were summarised using risk differences on the full population and on select subpopulations. These were estimated using augmented inverse propensity weighting (AIPW). AIPW is an asymptotically optimal way of estimating the RDs. Two-sided 95% CIs were estimated using asymptotic normality of the AIPW estimator.

CF was used to identify risk groups by considering interactions between important covariates. Important covariates were considered to be those with most splits in the CF. This was done because splits in causal trees are chosen to maximise the difference in CRD, so the number of splits can be used to uncover which covariates influence RDs. We then considered each three-way interaction between the four most important covariates. Three-way interactions were chosen to avoid subgroups with insufficient data to summarise the CRDs using AIPW. The frequency of splits along each covariate was recorded at each depth of the trees, and the final measure was a weighted average of these frequencies. The weight function used was the reciprocal squared of the tree depth.

Identifying assumptions behind CF were checked by assessing the overlap of exposure groups using a propensity score density plot, and by assessing the balance of covariates across exposure groups using the absolute standardised mean difference <0.1 to indicate balance. Model calibration was evaluated by computing the best linear fit of the target estimate using the mean forest prediction as well as the differential forest prediction as regressors. The c-for-benefit was calculated to evaluate the discrimination performance of the model[22]. Treatment effect heterogeneity was evaluated using Rank-Weighted Average Treatment Effect (RATE) metrics. A RATE metric takes a treatment prioritisation rule and assesses the rules ability to prioritise individuals with the largest treatment effect. Specifically, we used the area under the targeting operator characteristic curve (AUTOC). AUTOC allows an asymptotically valid test for the hypothesis of treatment effect heterogeneity along the prioritisation rule to be constructed using the bootstrap. P-values for the test of RD heterogeneity along risk factors were estimated using the AUTOC metric with standard error estimated from 500 bootstrap replicates. Overall heterogeneity was assessed by splitting the study data into two random subsamples, predicting CATEs based on CF models trained on the opposite subsample, and using these as the prioritisation rule for the RATE metric.

Technical details on the causal forest method as well as RATE metrics can be found in Supplementary methods 1 and elsewhere [19,23–26].

All statistical analyses were carried out in R version 4.2.2[27]. The R-package *grf* version 2.2.1 was used for modelling and *ggplot2* version 3.4.2 for data visualisation [28,29].

**Sensitivity analyses**. We conducted two sensitivity analyses. First, we evaluated the impact of false RT-PCR test results. Second, we evaluated the impact of our choice of hyperparameters in the causal forest algorithm. Details are in Supplementary methods 2.

**Ethical approval**. This study was performed as a surveillance study as part of the governmental institution Statens Serum Institut's (SSI) advisory tasks for the Danish Ministry of Health. SSI's purpose is to monitor and fight the spread of disease in accordance with section 222 of the Danish Health Act. According to Danish law, national surveillance activities carried out by SSI do not require approval from an ethics committee.

This study used data from the EFTER-COVID survey. Participation in the survey was voluntary. The invitation letter to participants contained information about their rights under the Danish General Data Protection Regulation (rights to access data, rectification, deletion, restriction of processing and objection). After reading this information, it was considered informed consent if participants agreed and clicked on the link to fill in the questionnaires.

**Reporting summary**. Further information on research design is available in the Nature Portfolio Reporting Summary linked to this article.

## Results

**Cohort characteristics**. 294,035 persons were invited to the survey in the study period. 106,917 (36.4%) persons fully completed the questionnaire sent out 9 months after the test date. The respondents were more often female (61.9% vs. 51.7%), older (≥50 years: 54.5% vs. 23.4%) and had more comorbidities (Charlson score > 0: 13.8% vs. 9.2%). The distribution of persons invited to the survey stratified by respondent status (respondents, non-respondents) can be found in Table 1. After applying the remaining exclusion criteria, the cohort consisted of 88,818 individuals, of which 37,482 had had a confirmed SARS-CoV-2 infection. A flowchart showing persons excluded after receiving the questionnaire can be found in Fig. 1. The mean age in the study cohort was 45 years with standard error 14 years and 64.3% were female. 62.1% had some form of higher education, while 16.0% had vocational training. The most prevalent existing health conditions before test were high BMI (16.6%), depression (12.1%), high blood pressure (11.1%), and anxiety (8.4%). Overall, the test-positive and test-negative cohorts are similar across the participant characteristics (see Model performance). The test-positive cohort has a higher proportion of males (38.6% vs. 33.6%) and is on average 2.2 years younger (43.6 vs. 45.8 years). In terms of pre-existing health conditions, the test-positive cohort has a smaller proportion of chronic fatigue syndrome (1.3% vs. 1.7%) and chronic obstructive pulmonary disease (1.1% vs. 1.5%), a larger proportion of chronic asthma (7.6% vs. 6.8%), and similar proportions of fibromyalgia (0.9% vs. 0.9%) and post-traumatic stress disorder (2.0% vs. 1.9%). All participant

characteristics by exposure group can be found in Supplementary Table 1.

**Distribution of sick leave**. From the study cohort, 7955 (9.0%) reported taking some amount of full-time sick leave >4 weeks after their test date. Of these, 2412 (30.3%) took substantial sick leave of >4 weeks. Shorter durations of sick leaves were reported more often than long durations. 346 people reported being on full-time sick leave for >9 months or since their test date. Test-positives reported more substantial full-time sick leave than test-negatives (4.5% of test-positives vs. 1.4% of test-negatives). Stratifying by sex, females reported full-time sick leave of any duration more often than males (9.7% of females vs. 7.6% of males). Similarly, individuals 50 years or older at the time of their test reported taking more full-time sick leave of any duration than

individuals below 50 years at the time of their test (9.7% of ≥50 years vs 8.3% of <50 years). The distribution of the duration of self-reported full-time sick leave taken >4 weeks after the test date can be found in Table 2.

**Risk differences**. CRD estimates from the CF model ranged from −3.4% to 13.7%. The RD by deciles of CRD increases from 1.3% (95% CI 0.8% to 1.8%) in the lowest decile to 5.4% (95% CI 4.2% to 6.6%) in the highest decile. The RD by deciles of CRD and the density of CRDs are shown in Fig. 2. 66.3% of CRDs were between 0% and 5%. Of the remaining CRDs, a majority (26.0%) were above 5%, while the remaining 7.7% were below 0%. Summarising the CRDs over the study population using AIPW showed persons infected with SARS-CoV-2 had a higher risk (RD 3.3%, 95% CI 3.1% to 3.6%) of taking substantial full-time sick leave after their acute infection compared to test-negatives with no known history of SARS-CoV-2 infection. This agrees with a previous analysis of the data, where the RD was estimated with parametric g-computation using a logistic regression model to predict outcomes on the two exposure groups[12]. AIPW was used to summarise the CRDs from the CF model in subpopulations defined by single risk factors. This yielded RDs within a few tenths of a percentage point of the g-computation estimates. The RD estimates are shown in Table 3.

**Risk groups**. According to the CF variable importance measure, the most important risk factors, in terms of maximising heterogeneity, were age (0.570), high BMI (0.212), depression (0.053), sex (0.037), education level (0.035), and chronic asthma (0.020) (Supplementary Fig. 2).

Results for each three-way interaction between age, high BMI, depression, and sex are shown in Figs. 3, 4, Supplementary Fig. 3, and Supplementary Fig. 4. Subgroup counts for combinations of age, high BMI, and depression are shown in Supplementary Table 2, while counts for combinations of sex, depression, and high BMI are shown in Supplementary table 3. Overall, the CRD increased with age, high BMI, and depression. CRDs were higher and varied more for females than males.

The interaction between age, high BMI, and depression show an increase in CRDs with each of these covariates (Fig. 3). The effect of SARS-CoV-2 infection on post-acute full-time sick leave

| Table 1 Characteristics of non-responders. | | | |
|---|---|---|---|
| **Characteristics** | **Complete** | **Partially complete** | **Incomplete** |
| | (n = 106,917) | (n = 10,928) | (n = 176,190) |
| **Sex** | | | |
| Female | 66,161 (61.9%) | 7004 (64.1%) | 91,057 (51.7%) |
| Male | 40,755 (38.1%) | 3924 (35.9%) | 85,131 (48.3%) |
| **Age (10-year categories)** | | | |
| 15–19 years | 4071 (3.8 %) | 1261 (11.5%) | 22,922 (13.0%) |
| 20–29 years | 13,000 (12.2%) | 2073 (19.0%) | 46,545 (26.4%) |
| 30–39 years | 12,647 (11.8%) | 1781 (16.3%) | 33,914 (19.2%) |
| 40–49 years | 18,969 (17.7%) | 1861 (17.0%) | 31,569 (17.9%) |
| 50–59 years | 27,014 (25.3%) | 1917 (17.5%) | 23,935 (13.6%) |
| 60–69 years | 19,551 (18.3%) | 1134 (10.4%) | 10,364 (5.9%) |
| 70+ years | 11,664 (10.9%) | 901 (8.2%) | 6939 (3.9%) |
| **Charlson Comorbidity Scores** | | | |
| 0 | 92,179 (86.2%) | 9431 (86.3%) | 159,920 (90.8%) |
| 1 | 7243 (6.8%) | 760 (7.0%) | 9163 (5.2%) |
| 2 | 5293 (5.0%) | 460 (4.2%) | 4750 (2.7%) |
| 3 or more | 2201 (2.1%) | 277 (2.5%) | 2355 (1.3%) |

Characteristics of n = 294,035 individuals who were sent a questionnaire 9 months after their test date and either completed, partially completed, or didn't complete the questionnaire. Information was missing for six individuals.

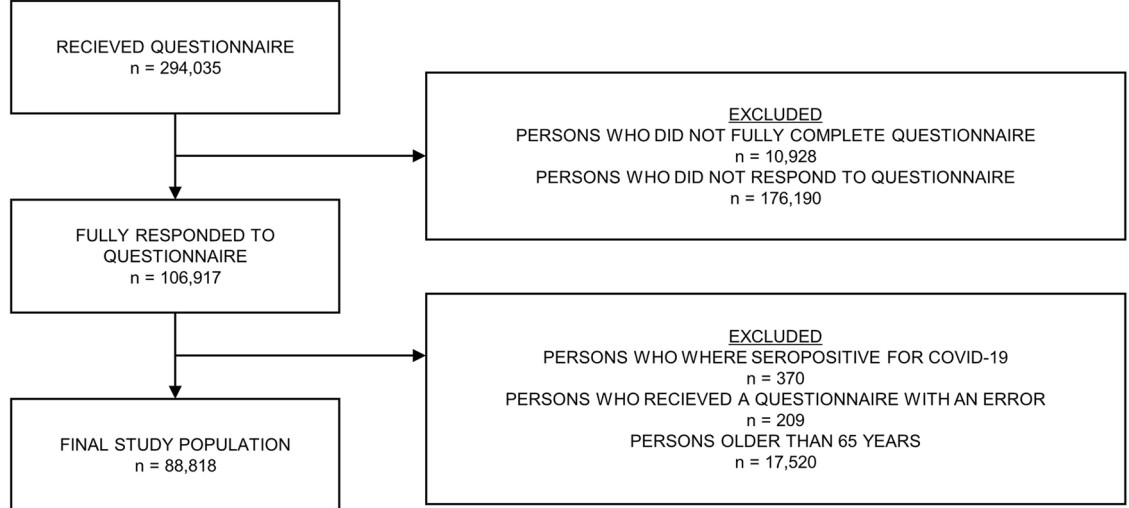

**Fig. 1 Diagram illustrating the flow of persons from receiving their questionnaire to inclusion in the final study population.** The flow of persons from receiving their questionnaire (n = 294,035) to inclusion in the final study populations (n = 88,818). Persons were exluded if they did not respond fully to their questionnaire. Of those that filled out the questionnaire (n = 106,117), persons above 65 years of age, persons with a seropositive result for COVID-19, and persons who received a questionnaire with an error were excluded.

**Table 2 Distribution of full-time sick leave.**

| Full-time sick leave | Full cohort | Female | Male | <50 years | ≥50 years |
|---|---|---|---|---|---|
| | n (%) | n (%) | n (%) | n (%) | n (%) |
| No sick leave | 80,863 (91.0) | 51,542 (90.3) | 29,321 (92.4) | 44,356 (91.7) | 36,507 (90.3) |
| <2 weeks | 4216 (4.8) | 2875 (5.0) | 1341 (4.2) | 2248 (4.7) | 1968 (4.9) |
| 2–4 weeks | 1327 (1.5) | 918 (1.6) | 409 (1.3) | 619 (1.3) | 708 (1.8) |
| 1–2 months | 1070 (1.2) | 788 (1.4) | 282 (0.9) | 525 (1.1) | 545 (1.4) |
| 2–4 months | 567 (0.6) | 413 (0.7) | 154 (0.5) | 263 (0.5) | 304 (0.8) |
| 4–6 months | 260 (0.3) | 184 (0.3) | 76 (0.2) | 126 (0.3) | 134 (0.3) |
| 6–9 months | 169 (0.2) | 123 (0.2) | 46 (0.1) | 88 (0.2) | 81 (0.2) |
| >9 months | 120 (0.1) | 81 (0.1) | 39 (0.1) | 46 (0.1) | 74 (0.2) |
| Since test date | 226 (0.3) | 163 (0.3) | 63 (0.2) | 108 (0.2) | 118 (0.3) |

| | Test negative | Test positive |
|---|---|---|
| | n (%) | n (%) |
| No sick leave | 47285 (92.1) | 33578 (89.6) |
| <2 weeks | 2972 (5.8) | 1244 (3.3) |
| 2–4 weeks | 347 (0.7) | 980 (2.6) |
| 1–2 months | 241 (0.5) | 829 (2.2) |
| 2–4 months | 192 (0.4) | 375 (1.0) |
| 4–6 months | 94 (0.2) | 166 (0.4) |
| 6–9 months | 72 (0.1) | 97 (0.3) |
| >9 months | 64 (0.1) | 56 (0.1) |
| Since test date | 69 (0.1) | 157 (0.4) |

Distribution of the amount of full-time sick leave reported by the study cohort >4 weeks after their test date.

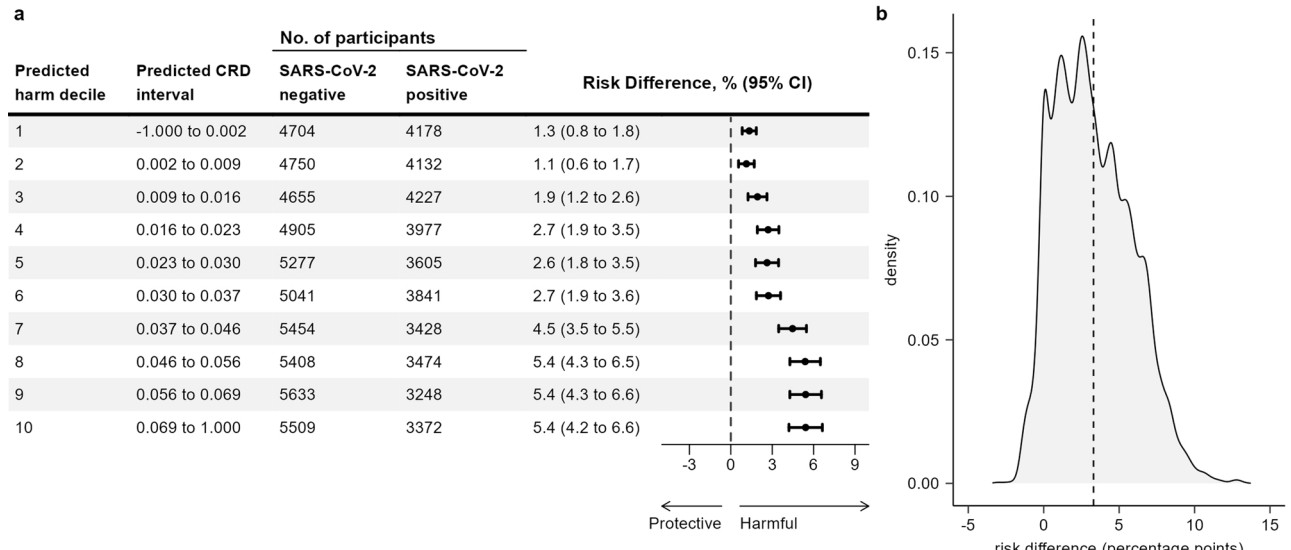

**Fig. 2 Heterogeneity of exposure effect estimated using out-of-bag prediction. a** Heterogeneity in the effect of COVID-19 exposure on substantial post-acute full-time sick leave by deciles of estimated conditional risk difference (CRD). Observations are ranked in ten separate folds, using a causal forest model with observations clustered by folds. Risk differences (RD) are estimated using augmented inverse propensity weighting (AIPW). Error bars show two-sided 95% confidence intervals. **b** Distribution of conditional risk differences from $n = 88,818$ individuals, estimated using out-of-bag prediction from the causal forest model. The dashed vertical line shows the average risk difference.

is smallest for persons below 50 years with no depression and no high BMI (mean 1.7%, SE 2.0%). 44.7% in the subgroup had CRDs significantly below the average RD for the full population (z-test with two-sided alternative and 5% significance level). The variance was smallest for persons above 50 years with depression and high BMI. This group also had the largest CRDs (mean 7.5%, SE 1.7%). 99.7% of persons in the subgroup had CRDs above the average RD for the full population, while 49.6% were significantly above. The effect of SARS-CoV-2 infection on substantial post-acute full-time sick leave was similar for persons with either depression or high BMI and increased compared to having

neither depression nor high BMI. For persons below 50 years, the CRD distribution for persons with depression had mean 3.4% and SE 2.2%, while for persons with high BMI, the mean was 4.4% and SE was 2.1%. For persons above 50 years, the CRD distribution for persons with depression had mean 5.6% and SE 1.8%, while for persons with high BMI, the mean was 5.5% and SE was 2.1%. The RD estimate was smallest for persons aged 15–25 years with neither depression nor high BMI (0.4%, 95% CI 0.1% to 0.8%), while the RD was largest for persons aged 36–45 years with both depression and high BMI (10.4%, 95% CI 5.1% to 15.6%). The RD was mostly constant from age

**Table 3 Causal forest risk differences, RATE, and variable importance.**

|  | n | RD (95% CI) | RATE (SE) | RATE P-value | variable importance |
|---|---|---|---|---|---|
| Full population | 88,818 | 3.3 (3.1 to 3.6) | 9.4 (1.4) | $6.6 \cdot 10^{-14}$ |  |
| Female | 57,087 | 3.9 (3.6 to 4.3) | −6.5 (0.9) | $1.6 \cdot 10^{-12a}$ | 0.037[a] |
| Male | 31,731 | 2.2 (1.8 to 2.5) |  |  |  |
| Older than 50 years | 40,439 | 4.3 (3.8 to 4.7) | 10.5 (1.3) | $1.2 \cdot 10^{-14b}$ | 0.570[b] |
| Fibromyalgia | 803 | 10.3 (6.4 to 14.1) | 3.0 (0.9) | $2.1 \cdot 10^{-4}$ | 0.007 |
| COPD or other lung disease | 1197 | 7.0 (3.9 to 10.0) | 2.2 (0.9) | 0.016 | 0.004 |
| Diabetes | 2980 | 6.4 (4.6 to 8.2) | 3.6 (1.1) | $5.4 \cdot 10^{-4}$ | 0.007 |
| Post-traumatic stress disorder | 1757 | 5.8 (3.4 to 8.2) | 2.0 (1.0) | 0.036 | 0.013 |
| High BMI | 14,700 | 5.6 (4.9 to 6.4) | 9.2 (1.3) | $3.9 \cdot 10^{-12}$ | 0.212 |
| Depression | 10,715 | 5.7 (4.8 to 6.6) | 7.0 (1.3) | $5.1 \cdot 10^{-8}$ | 0.053 |
| Chronic asthma | 6335 | 5.4 (4.2 to 6.5) | 4.2 (1.2) | $1.4 \cdot 10^{-4}$ | 0.020 |
| High blood pressure | 9817 | 5.0 (4.1 to 5.9) | 4.8 (1.2) | $7.3 \cdot 10^{-5}$ | 0.004 |
| Chronic or frequent headaches | 3593 | 4.9 (3.3 to 6.6) | 2.2 (1.1) | 0.040 | 0.012 |
| Chronic fatigue syndrome | 1345 | 3.3 (0.7 to 5.9) | −0.03 (0.8) | 0.975 | 0.004 |
| Anxiety | 7425 | 3.4 (2.4 to 4.4) | 0.3 (1.1) | 0.809 | 0.007 |

Risk differences (RDs) and two-sided 95% confidence intervals (CI) obtained using CF and AIPW for full-time sick leave taken 4 weeks to 9 months after the test date between SARS-CoV-2 test-positives and test-negatives for the total study population and possible PCC risk groups. The RATE along with P-values for a RATE based test of treatment effect heterogeneity are reported along each risk factor. A RATE based omnibus test of treatment effect heterogeneity is also provided. All RATEs in the table have been multiplied by 1000.
[a]The RATE P-value tests for a difference in RDs between females and males. The variable importance scores split on sex.
[b]The RATE P-value tests for heterogeneity along the continuous age. The variable importance scores split on the continuous age variable.

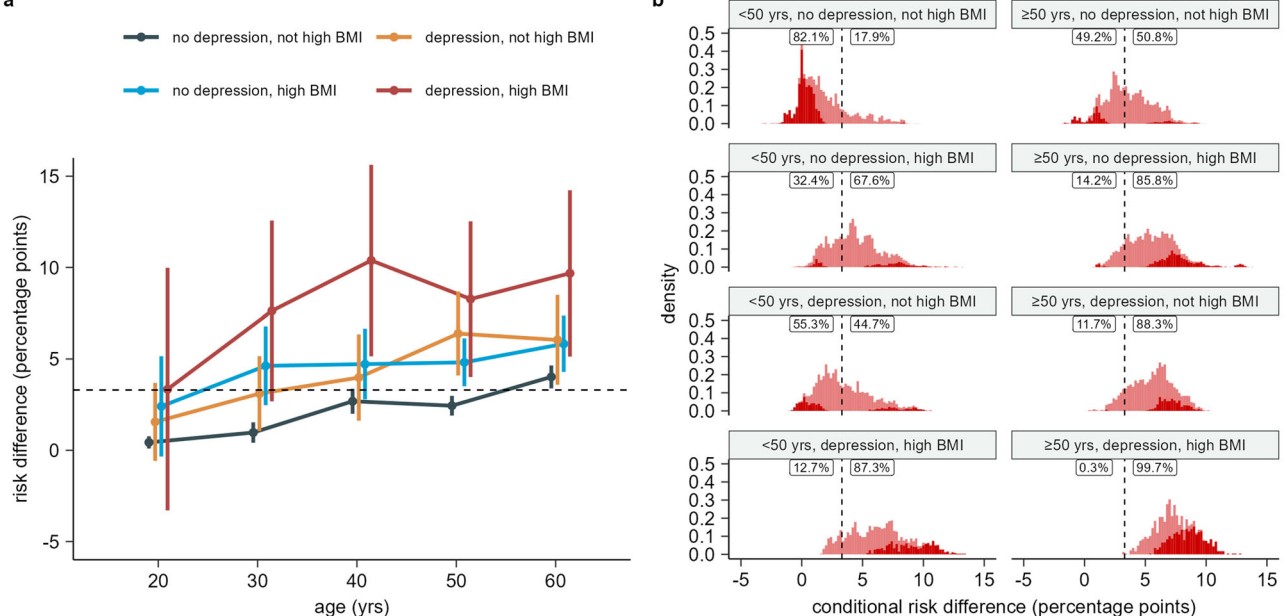

**Fig. 3 Estimated risk differences for combinations of age, high BMI, and depression. a** Risk differences (RDs) and two-sided 95% confidence intervals (CI) for substantial full-time sick leave taken 4 weeks to 9 months after the test date between SARS-CoV-2 test-positives and test-negatives for subgroups of a population of $n = 88,818$ individuals. Subgroups are defined by combinations of age, high BMI, and depression risk factors. Subgroups with unknown BMI ($n = 6823$) are not displayed. **b** Distribution of conditional risk differences (CRDs), estimated using out-of-bag prediction from the causal forest model, within subgroups of a population of $n = 88,818$ individuals. Subgroups are defined by combinations of age, high BMI, and depression risk factors. Subgroups with unknown BMI ($n = 6823$) are not displayed. The dashed line shows the average RD in the full population. The fraction of CRDs below and above this RD is printed for each combination of age, high BMI, and depression. Dark red indicates estimated CRDs significantly different from the average RD in the full population at a 5% significance level using a z-test with two-sided alternative.

36–65 years, with an increasing trend for persons aged 15–35 (Fig. 3). Combining the older age groups from 36–65 years, we found a RD of 9.3% (95% CI 6.6% to 12.0%).

The interaction between sex, depression, and high BMI show higher and more varied CRDs for females (Fig. 4). Among males with neither depression nor high BMI, the CRD distribution had mean 1.8% and SE 1.7%, while for females, the mean was 3.0% and the SE was 2.3%. Among males, 40.4% had CRDs significantly below the RD for the full population. Among females, this number was 22.1%. For females, we observed a

smaller increase in CRDs with depression compared with high BMI. The average CRD increased from 3.0% to 4.7% and 5.5% respectively. The RD estimates showed no difference between depression and high BMI for females. With depression, the RD was 5.6% (95% CI 4.3% to 6.9%), and with high BMI the RD was 5.5% (95% CI 4.4% to 6.6%). For males, the average CRD also increased less for persons with depression compared with high BMI, with an increase in average CRD from 1.8% to 2.9% and 4.4% respectively. However, the RD estimates also show a significant difference (two sample t-test with two-sided

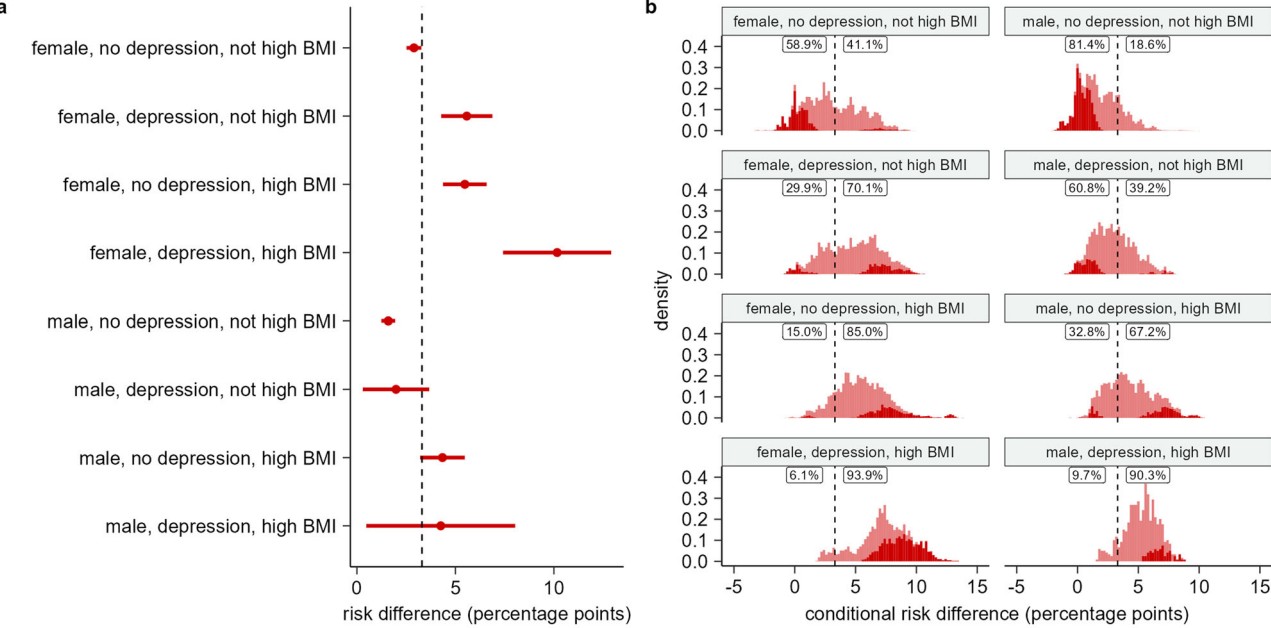

**Fig. 4 Estimated risk differences for combinations of sex, depression, and high BMI. a** Risk differences (RDs) and two-sided 95% confidence intervals (CI) for substantial full-time sick leave taken 4 weeks to 9 months after the test date between SARS-CoV-2 test-positives and test-negatives for subgroups of a population of $n = 88{,}818$ individuals. Subgroups are defined by combinations of sex, depression, and high BMI risk factors. Subgroups with unknown BMI ($n = 6823$) are not displayed. **b** Distribution of conditional risk differences, estimated using out-of-bag prediction from the causal forest model, within subgroups of a population of $n = 88{,}818$ individuals. Subgroups are defined by combinations of sex, depression, and high BMI risk factors. Subgroups with unknown BMI ($n = 6823$) are not displayed. The dashed line shows the average RD in the full population. The fraction of CRDs below and above this RD is printed for each combination of sex, depression, and high BMI. Dark red indicates estimated CRDs significantly different from the average RD in the full population at a 5% significance level using a $z$-test with two-sided alternative.

alternative, $t = -2.3$, df = 3852, $P$-value 0.023). The RD for males with depression and without high BMI was 2.0% (95% CI 0.3% to 3.7%), while for males without depression and with high BMI, the RD was 4.3% (95% CI 3.2% to 5.5%). Among persons with both depression and high BMI, the CRDs were again higher and more varied for females. They also had the highest RD (10.2%, 95% CI 7.4% to 12.9%).

A secondary analysis looking at more three-way interactions with age, sex, and additional health conditions can be found in Supplementary Results 1 and Supplementary Fig. 5.

**Model performance**. The predicted propensity scores used by the CF showed no evidence of positivity violations. They were concentrated away from 0 and 1, with good overlap between the distributions for test negatives and -positives (Supplementary Fig. 6). The unadjusted imbalance of the covariates across exposure groups was low. The only covariate above a 0.1 absolute standardised mean difference was age. After weighting with the inverse propensity, the overall imbalance was considerably reduced, and all covariates achieved good balance (Supplementary Table 4, Supplementary Figs. 7–10). In the best linear fit model for the target estimand, the coefficient of the mean forest prediction was 1.00 (95% CI 0.93 to 1.07), indicating that the mean forest prediction was well calibrated. The coefficient of the differential forest prediction was 0.66 (95% CI 0.56 to 0.76), indicating a potential lack of calibration of the estimated treatment heterogeneity. The c-for-benefit of the causal forest model was 0.59 (95% CI 0.58 to 0.60), indicating reasonable discrimination performance of the model[22]. The AUTOC test of heterogeneity showed strong evidence for variability in RDs within the total study population as well as most risk groups defined by single risk factors, apart from anxiety ($P$-value 0.8) and chronic fatigue syndrome ($P$-value 1.0) (Table 3).

**Sensitivity analyses**. In a sensitivity analysis evaluating the impact of false RT-PCR test results, the variable importance measure consistently ranked the same five risk factors highest as in the main analysis, see Supplementary Fig. 11. Risk differences were slightly lower than in the main analysis, but evidence for heterogeneity remained strong, see Table 4 and Fig. 5.

In a sensitivity analysis evaluating the impact of our choice of hyperparameters in the causal forest algorithm, calibration of the heterogeneity estimates depended on the choice of hyperparameters. Results on variable importance were similar to the RT-PCR sensitivity analysis, see Supplementary Fig. 12. Overall, there was no evidence that the estimated RDs were sensitive to the choice of hyperparameters in the causal forest algorithm, excluding parameters controlling the imbalance between child leaves in a split, which were kept fixed, see Table 5 and Fig. 5.

Detailed sensitivity analysis results can be found in Supplementary Results 2.

## Discussion

The interplay between individual-level risk factors and post COVID-19 condition is complex and multifaceted and may not be fully understood by examining single risk factors in isolation. As such, it is important to model the interactions of various health characteristics in order to identify core risk factors, to understand the impact of pre-existing multimorbidity on PCCs, and to inform appropriate interventions. We built upon previous work, which observed a higher risk of substantial post-acute sick leave after infection with SARS-CoV-2 during the index and alpha waves[12]. Here, we identified previously unknown subgroups with high increased risk of substantial post-acute full-time sick leave after COVID-19 infection. Using a state-of-the-art causal machine learning algorithm and a causal forest variable importance measure, we identified age, high BMI, depression, and

**Table 4 RT-PCR test sensitivity analysis—estimated risk differences for single risk factors.**

|  | n | E(RD) (min, max) | E(RATE) (min, max) | largest RATE P-value |
|---|---|---|---|---|
| Full population | 88,818 | 2.9 (2.8 to 3.0) | 9.7 (8.0, 11.0) | 5.4 · 10⁻¹⁰ |
| Female | 57,087 | 3.4 (3.4 to 3.6) | −5.5 (−6.5, −5.0) | 7.5 · 10⁻⁹ᵃ |
| Male | 31,731 | 1.9 (1.8 to 2.0) |  |  |
| Older than 50 years | 40,439 | 3.7 (3.6 to 3.8) | 8.8 (8.3, 9.7) | 2.0 · 10⁻¹⁰ᵇ |
| Fibromyalgia | 803 | 9.1 (8.5 to 9.8) | 2.6 (2.4, 3.0) | 0.003 |
| COPD or other lung disease | 1197 | 5.8 (5.2 to 6.7) | 1.7 (1.4, 2.2) | 0.089 |
| Diabetes | 2980 | 5.6 (5.0 to 6.2) | 3.2 (2.6, 3.9) | 0.010 |
| Post-traumatic stress disorder | 1757 | 5.0 (4.2 to 5.9) | 1.6 (1.0, 2.4) | 0.253 |
| High BMI | 14,700 | 5.0 (4.6 to 5.2) | 8.3 (7.2, 8.9) | 2.7 · 10⁻⁹ |
| Depression | 10,715 | 5.0 (4.6 to 5.3) | 6.0 (4.8, 7.1) | 3.6 · 10⁻⁵ |
| Chronic asthma | 6335 | 4.7 (4.3 to 5.0) | 3.6 (2.9, 4.1) | 0.005 |
| High blood pressure | 9817 | 4.5 (4.1 to 4.9) | 4.3 (3.2, 5.4) | 0.006 |
| Chronic or frequent headaches | 3593 | 4.4 (3.6 to 5.1) | 2.0 (1.0, 2.9) | 0.341 |
| Chronic fatigue syndrome | 1345 | 2.8 (1.7 to 4.1) | −0.1 (−0.7, 0.7) | 0.986 |
| Anxiety | 7425 | 3.2 (2.6 to 3.2) | −0.1 (−0.6, 0.6) | 0.996 |

Results of sensitivity analysis where false negatives and false positives are sampled from test negatives and test positives and moved to the correct exposure group. Results are based on n = 20 simulations assuming a sensitivity of 90% and a specificity of 99%. Results were obtained using CF and AIPW for substantial full-time sick leave taken 4 weeks to 9 months after the test date between SARS-CoV-2 test-positives and test-negatives for the total study population and possible PCC risk groups. All RATEs in the table have been multiplied by 1000.
ᵃ The RATE P-value tests for a difference in RDs between females and males. The variable importance scores split on sex.
ᵇThe RATE P-value tests for heterogeneity along the continuous age. The variable importance scores split on the continuous age variable.

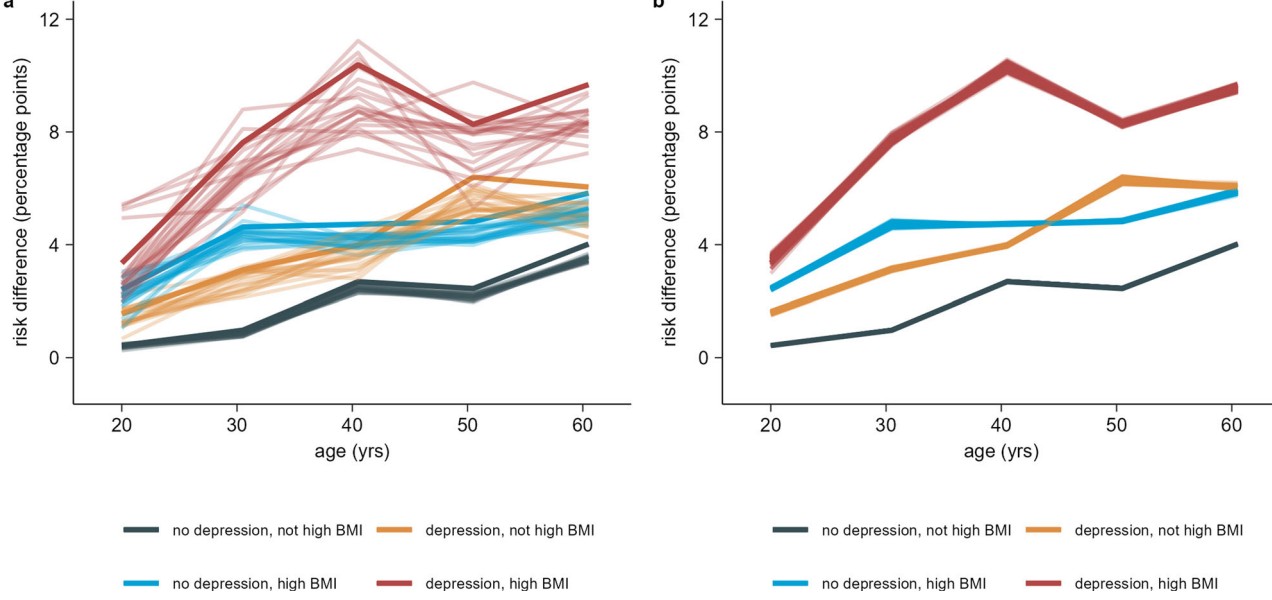

**Fig. 5 Sensitivity analyses - Estimated risk differences for combinations of age, high BMI, and depression. a** Risk differences (RDs) for substantial full-time sick leave taken 4 weeks to 9 months after the test date between persons labelled as true positive or false negative, and persons labelled as true negative or false positive. Estimates are across subgroups of a population of n = 88,818 individuals. Subgroups are defined by combinations of age, high BMI, and depression risk factors. Subgroups with unknown BMI (n = 6823) are not displayed. The solid coloured lines show the RDs from the main analysis, while the transparent lines show the result from each analysis using a modified exposure. **b** Risk differences (RDs) for substantial full-time sick leave taken 4 weeks to 9 months after the test date between SARS-CoV-2 test-positives and test-negatives within subgroups of a population of n = 88,818 individuals. Subgroups are defined by combinations of age, high BMI, and depression risk factors. Subgroups with unknown BMI (n = 6823) are not displayed. The plot consists of 81 lines for each combination of depression and high BMI, each showing the result for a different combination of the hyperparameters sample fraction, mtry, minimum node size, and honesty fraction.

sex as the most important variables for explaining observed effect modification on post-acute sick leave following SARS-CoV-2 infection.

We observed a high amount of effect heterogeneity, with significant modification for all considered risk factors except anxiety and chronic fatigue syndrome. We also identified subgroups defined by three-way risk factor interactions with high risk difference estimates. Among the three-way interactions considered, females with high BMI and depression as well as persons aged 36–45 years with high BMI and depression had a large increase in

their risk of substantial post-acute sick leave above 10%. The estimated risk difference was also high for older individuals with high BMI and depression, and we found the combined RD for those aged older than 35 to be 9.3%. False RT-PCR test results due to imperfect sensitivity and specificity may have caused a small overestimation of the increased risk.

These results build on risk factors previously found to increase sick leave prevalence. A German study by Jacob et al. reported an overall prevalence of long-term COVID-19 sick leave of 5.8%, and a significant increase in prevalence for female sex (6.5%),

**Table 5 Hyperparameter sensitivity analysis—estimated risk differences for single risk factors.**

|  | n | Risk difference | | |
|---|---|---|---|---|
|  |  | Average | Minimum | Maximum |
| Full population | 88,818 | 3.31 | 3.29 | 3.32 |
| Female | 57,087 | 3.94 | 3.93 | 3.95 |
| Male | 31,731 | 2.18 | 2.15 | 2.20 |
| Older than 50 years | 40,439 | 4.27 | 4.24 | 4.29 |
| Fibromyalgia | 803 | 10.36 | 10.18 | 10.54 |
| COPD or other lung disease | 1197 | 6.89 | 6.79 | 7.04 |
| Diabetes | 2980 | 6.35 | 6.27 | 6.49 |
| Post-traumatic stress disorder | 1757 | 5.79 | 5.69 | 5.87 |
| High BMI | 14,700 | 5.66 | 5.60 | 5.69 |
| Depression | 10,715 | 5.65 | 5.59 | 5.71 |
| Chronic asthma | 6335 | 5.40 | 5.27 | 5.46 |
| High blood pressure | 9817 | 5.00 | 4.98 | 5.04 |
| Chronic or frequent headaches | 3593 | 4.99 | 4.86 | 5.05 |
| Chronic fatigue syndrome | 1345 | 3.24 | 3.10 | 3.32 |
| Anxiety | 7425 | 3.40 | 3.34 | 3.47 |

The average, minimum, and maximum risk difference for the full population and different single risk factors obtained from a sensitivity analysis repeating the main analysis with different combinations of the hyperparameters sample fraction, mtry, minimum node size, and honesty fraction used by the causal_forest function from the grf R package. Three values of each hyperparameter was used, for a total of 81 configurations tested. Results were obtained using CF and AIPW for substantial full-time sick leave taken 4 weeks to 9 months after the test date between SARS-CoV-2 test-positives and test-negatives for the total study population and possible PCC risk groups.

older age (56–65 years: 10.7%), obesity (9.1%), and depression (8.6%), among other factors[11]. Colleagues previously reported significant increases in RDs for these factors using g-computation on our study cohort[12]. In the present study, our results suggest that having multiple risk factors preceding COVID-19 infection could further increase the effect of infection on the risk of post-acute full-time sick leave.

The transportability of our findings to other countries' populations may be most relevant when considering how having a combination of risk factors (specifically older age, high BMI, depression, and female sex) can collectively impact one's health, even after the acute COVID-19 infection. While we were able to examine the burden with respect to post-acute sick leave, other countries may be able characterise this burden in other ways, such as by examining differences in employment termination or disability payments.

Naturally, this study has its limitations. First, our cohort consists of individuals who tested positive between November 2020 and February 2021, a period dominated by the index and alpha variants and before any large-scale vaccine roll-out in Denmark. Today, vaccine coverage is high[30], and newer (omicron) variants are dominant. Both factors are associated with a reduction in PCC prevalence[14,31,32], raising the issue of how to interpret the results in the current pandemic context.

Second, there are some limitations with our choice of outcome. One problem is that sick leave only measures those with a connection to the labour marked. We note that this includes students and the unemployed. In Denmark, persons who are unemployed can only receive unemployment benefits if they are actively looking for a job. If a person is unable to comply with their obligations due to illness or injury, they can apply for sick leave[33]. Similarly, students can apply for sick leave if illness or injury prevents them from attending their exams. Due to the design of the EFTER-COVID questionnaire we cannot rule out the presence of persons ineligible for the outcome in the study cohort, such as persons on early retirement. We presume these persons will not report taking sick leave, since they have no incentive for false reporting, although some could have misinterpreted the question. Overall, we expect the vast majority of the study population to be eligible for the outcome.

Third, prior full-time sick leave could be a good predictor of post-acute full-time sick leave, but due to participants in the EFTER-COVID survey not being asked about it, we are unable to adjust for this covariate. If prior full-time sick-leave is associated with the test result, this will introduce bias in our results. Participants on full-time sick leave prior to their test-date may to some extent be captured by the Charlson comorbidity index.

Furthermore, the most likely association would be that prior sick leave was associated with a lower probability of testing positive due to decreased social activities. Since persons already on sick-leave are more likely to report substantial sick leave after infection, these assumptions imply that our reported risk difference estimates are conservative compared to what we would get if we could adjust for prior sick-leave, since people with high risk of long-term sick leave end up overrepresented in the test-negative group.

Fourth, there are limitations from using the EFTER-COVID questionnaire data. One concern is the self-reporting nature of the questionnaire, which is vulnerable to reporting errors. In particular, there is potential for recall bias, where participants may not remember the amount of sick leave they took over the 9 months following the test. Further, there is potential for selection bias due to applied exclusion criteria resulting in a non-representative study population. First, only tested individuals were eligible for invitation to the EFTER-COVID questionnaire. However, we believe this to be a minor concern due to the universal testing strategy for SARS-CoV-2 implemented in Denmark from May 2020 through the study period, where testing was encouraged and freely available to all adults[20,34]. Second, just over 1 in 3 of the invitees completed the questionnaire. We observed that respondents were more likely female, older, and with more comorbidities compared with non-respondents. As these groups had higher than average risk differences, it is likely that the results reported in Table 3 are exaggerated compared with the average risk difference in the general Danish working age population. It is also possible participants experiencing post-acute symptoms and taking sick-leave were more motivated to respond to the questionnaire, leading to exaggerated RD estimates. If this was the case, we should expect a higher response rate among test positives, since we observed an increased risk of substantial post-acute

full-time sick leave after a positive test compared with a negative test (RD 3.3%).

Finally, from a methodological perspective, concerns have been cited about inconsistency in variables found to impact heterogeneity across different supervised machine learning methods, as well as across random seeds [35].

We found the parameter tuning procedure implemented by the *grf* package using cross-validation would produce unstable and sometimes ill-calibrated results. This was at least partly caused by hyperparameters in the causal forest algorithm that control the maximum imbalance allowed between two child leaves in a split. Due to the low prevalence of several of the health conditions considered, we found it was important to adjust these parameters to allow splitting along low prevalence health conditions. When using the parameter tuning procedure, the output forest was quite restrictive on the maximum imbalance. When splits are not permitted due to large imbalance, their variable importance will be 0, and CRDs are constant along the low prevalence health conditions, irrespective if heterogeneity exists. Controlling the imbalance is done to align the algorithm with theory[29]. However, not considering the effect of this can lead to potentially important heterogeneity being ignored. For our cohort, fibromyalgia and COPD end up without splits unless we adjust the imbalance parameters.

We ended up using the default values for the tuning parameters used by the causal forest algorithm, but with modification to the parameters controlling split imbalance and maximum node size. By performing a sensitivity analysis, we found causal forest produced consistent results regarding variables found to impact heterogeneity across different combinations of hyperparameters not controlling split imbalance.

These results do not address the concern about inconsistency across supervised learning methods. It would be valuable to repeat our analysis with an alternative method; however, we are unaware of any alternative accessible methods to replace our causal forest analysis with the goal of identifying subgroups with high causal risk differences and providing patient-centred estimates of excess risk that take multimorbidity into account.

Additional limitations are discussed in the Supplementary discussion.

Altogether, our study employs causal machine learning to investigate heterogeneity in the effect of COVID-19 on post-acute sick leave and to demonstrate the value of individual level causal effect estimates to identify persons particularly susceptible to the burden of PCC. The results highlight the potential for targeted public health interventions that account for individual PCC risk, with age, sex, high BMI, and depression identified as key factors, and serves as an example of the use of the causal forest approach in the observational setting.

## Data availability

The datasets used in this study comprise sensitive, individual-level information from completed questionnaires and national register data. According to the Danish data protection legislation, the authors are not permitted to share these sensitive data directly upon request, including source data for figures. However, the data are available for research purposes upon request to the Danish Health Authority (register data, email: kontakt@sundhedsdata.dk) and Statens Serum Institut (questionnaire data, email: aii@ssi.dk), as well as within the framework of the Danish data protection legislation and any required permission from authorities. Data request processing can take an expected three to 6 months.

## Code availability

The analysis code is stored in a GitHub repository at https://github.com/kjakobse/risk-factors-associated-with-long-term-sick-leave-following-COVID-19-in-Danish-population[36].

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

## Acknowledgements

The study was funded by a Novo Nordisk Foundation Data Science Investigator grant awarded to AH. The funder had no role in the study design, data collection, analysis, interpretation of results, writing of the manuscript, or decision to submit the paper for publication. The authors would like to thank the EFTER-COVID research group from Statens Serum Institut comprising of Anna Irene Vedel Sørensen, Anders Koch, Jørgen Vinsløv Hansen, Lampros Spiliopoulos, Nete Munk Nielsen, Peter Bager, Poul Videbech, and Steen Ethelberg.

## Author contributions

A.H. and K.J. conceptualised this study. A.H. and K.J. designed the methodology. K.J. and E.O. reviewed the literature. I.S. prepared the data and K.J. validated it. K.J. did the data analysis, including code writing. K.J. and E.O. prepared the original draft of the manuscript. All authors have critically revised the manuscript. A.H. supervised the study. A.H. acquired funding. All authors have approved the final version of the manuscript and agreed to be accountable for all aspects of the work. The decision to submit was made by all authors. The corresponding author attests that all listed authors meet authorship criteria and that no others meeting the criteria have been omitted. K.J. and A.H. are the guarantors.

## Competing interests

The authors declare no competing interests.
