## [Peer Review File · Communications Medicine]

Reviewers' comments:

Reviewer #1 (Remarks to the Author):

In this study, the authors investigated the heterogeneity in the effect of SARS-CoV-2 infection on long-term sick leave and aimed to identify subgroups at higher risk. They used a self-reported survey and a register-based cohort study from Denmark and employed state-of-the-art causal forest methods to identify subgroups with prolonged sick leave. The authors identified four risk factors that explain the highest level of heterogeneity: age, sex, high BMI, and depression. Overall, the study was well conducted and reported. I have a few comments to improve it.

1. In this study, the authors investigated the heterogeneity in the effect of SARS-CoV-2 infection on long-term sick leave and aimed to identify subgroups at higher risk. They used a self-reported survey and a register-based cohort study from Denmark and employed state-of-the-art causal forest methods to identify subgroups with prolonged sick leave. The authors identified four risk factors that explain the highest level of heterogeneity: age, sex, high BMI, and depression. Overall, the study was well conducted and reported. I have a few comments to improve it.
2. In summary, could you clarify what the "test-negative control group" refers to? Are these individuals with a negative RT-PCR test? I noticed this term in the methods section, but it would be helpful to provide this clarification in the abstract as well.
3. The introduction is nicely written and clearly addresses the gap and the need for data-driven approaches to identify subgroups with the highest risk of sick leave.
4. Out of 294,035 individuals, 36.4% completed the survey sent out after 9 months. Firstly, it is important to note that individuals who were very sick and died within 9 months or those who were infected within 9 months would have been excluded from this survey. Do you have information on the distribution of both these groups? Secondly, it would be worth discussing whether individuals who completed the survey after 9 months differ from those who did not complete the survey (responder bias). Both of these points deserve discussion in the limitations section, highlighting how they can potentially influence the results and conclusions.
5. In the results section (lines 71-78), could you clarify if the tested positive and negative cohorts are similar or different in terms of patient characteristics? You present their characteristics in Table S1 and report them in the results, but it is not explicitly mentioned whether both cohorts are similar or different, and for which characteristics.
6. Regarding Figure S2, could you provide two histograms overlaying each other, one for the tested positive group and one for the tested negative group? It would also be helpful to show the overlap before and after IPTW weighting.
7. In the methods and results sections, could you describe the distribution of sick leaves? For instance, how many individuals (n and %) took 4, 5, 6,... weeks of leave? Providing this descriptive distribution would be useful for the readers.
8. In the discussion section, it would be valuable to comment on the generalizability of these findings to other countries.
9. In the results section (minor point), please number the tables and figures consecutively. For example, mention Figure S5 and S6 after Figure S2, S3, and S4.

Reviewer #2 (Remarks to the Author):

In the paper "Identifying heterogeneity in the effect of COVID-19 on long-term sick leave using causal machine learning", the authors develop a model to predict the conditional risk difference for an individual to take a long-term sick leave after a SARS-CoV-2 infection compared with not being infected using methods of causal ML and study the heterogeneity of this risk. For that purpose, they rely on the data of a hybrid survey and a register-based cohort study. The paper may provide some tools to assess certain aspects of the socio-economic cost of long covid.

The paper is well-written and well-structured. I have several important methodological comments and remarks:

1/The inclusion/exclusion criteria of the study used are not reported, which should be corrected.

2/Partly due to the abovementioned issue, it is impossible to determine if all the subjects considered in the study are eligible for the outcome. For instance, it is unclear if unemployed persons or students are eligible for long-term sick leave and how they have been accounted for in the study. This should be discussed and it may be necessary to conduct a sensitivity analysis

3/There may be censoring regarding the length of the sick leave. For example, how has someone who has been on sick leave for three weeks (and is still on sick leave) at nine-month after its positive SARS-CoV-2 positive test been accounted for in the study?

4/The causal groups (infected/not infected) are not as well defined as they may seem. The RT-PCR tests have a particular specificity and sensitivity, resulting in false positive or false negative individuals. It would be interesting to conduct a sensitivity analysis to evaluate if this influences the results or, at the very least, discuss it.

5/It seems that the data do not include information regarding whether individuals had already been on long-term sick leave prior to their confirmed SARS-CoV-2 infection. Such a covariable could be an excellent predictor of post-SARS-CoV-2 long-term sick leave. This situation should at least be thoroughly discussed in the paper if it is impossible to adjust/conduct a sensitivity analysis on this covariable.

6/The authors use a causal forest to estimate heterogeneity, applying standard metrics to ensure it is well-tuned. They also indicate that the automatic method for choosing the hyperparameters is unstable. Therefore, it would be interesting for them to confirm their findings using another class of model. This is especially important since importance techniques (in this case, the number of splits along a given covariate which is specific to causal forests) are highly dependent on the model's quality. Inadequate models can lead to incorrectly declaring important a variable that is not important and reciprocally.

7/ There is a closing parenthesis missing on the left panel of Figure 2 around percentage points

8/Same comments for Figure S5.

Response to Referees

Referee #1:

In this study, the authors investigated the heterogeneity in the effect of SARS-CoV-2 infection on long-term sick leave and aimed to identify subgroups at higher risk. They used a self-reported survey and a register-based cohort study from Denmark and employed state-of-the-art causal forest methods to identify subgroups with prolonged sick leave. The authors identified four risk factors that explain the highest level of heterogeneity: age, sex, high BMI, and depression. Overall, the study was well conducted and reported. I have a few comments to improve it.

1. In this study, the authors investigated the heterogeneity in the effect of SARS-CoV-2 infection on long-term sick leave and aimed to identify subgroups at higher risk. They used a self-reported survey and a register-based cohort study from Denmark and employed state-of-the-art causal forest methods to identify subgroups with prolonged sick leave. The authors identified four risk factors that explain the highest level of heterogeneity: age, sex, high BMI, and depression. Overall, the study was well conducted and reported. I have a few comments to improve it.

Response: We thank reviewer 1 for their feedback and appreciate the opportunity to improve our manuscript in-line with their comments.

2. In summary, could you clarify what the "test-negative control group" refers to? Are these individuals with a negative RT-PCR test? I noticed this term in the methods section, but it would be helpful to provide this clarification in the abstract as well.

Response: A clarification of the test-negative control group has been added to the summary. These are persons who were tested negative for COVID-19 with a RT-PCR test within the study period (November 4, 2020 – February 1, 2021), and who did not have a positive RT-PCR test registered before being invited to participate in the survey.

3. The introduction is nicely written and clearly addresses the gap and the need for data-driven approaches to identify subgroups with the highest risk of sick leave.

Response: We are pleased to hear that this section of the paper came across clearly.

4. Out of 294,035 individuals, 36.4% completed the survey sent out after 9 months. Firstly, it is important to note that individuals who were very sick and died within 9 months or those who were infected within 9 months would have been excluded from this survey. Do you have information on the distribution of both these groups? Secondly, it would be worth discussing whether individuals who completed the survey after 9 months differ from those who did not complete the survey (responder bias). Both of these points deserve discussion in the limitations section, highlighting how they can potentially influence the results and conclusions.

Response: We have no person-level information on persons who died or were infected before 9 months after their test date. Deaths or emigration could potentially create a non-representative selection, as deaths are more prevalent among the older population, while emigration is more prevalent among the younger population.¹ However, only a small proportion of the total population died or emigrated in the study period (around 0.2% died in 2021 while 1.1% emigrated) so the potential for selection is limited. Similarly, we expect limited selection due to positive tests in the follow-up period. From November 1, 2020 – October 31 2021, around 344000 confirmed SARS-CoV-2 cases (approx. 6 % of the population) were registered in Denmark.

Information on persons who did not respond to the survey has been added to the manuscript in Table 1. This shows they are more likely male, younger, and with fewer comorbidities compared with persons who completed the survey. A discussion of the potential influence of this selection has been added to the discussion section (lines 249-257): “just over 1 in 3 of the invitees completed the questionnaire. We observed that respondents were more likely female, older, and with more comorbidities compared with non-respondents. As these groups had higher than average risk differences, it is likely that the results reported in Table 3 are exaggerated compared with the average risk difference in the general Danish working age population. It is also possible participants experiencing post-acute symptoms and taking sick-leave were more motivated to respond to the questionnaire, leading to exaggerated RD estimates. If this was the case, we should expect a higher response rate among test positives, since we observed an increased risk of substantial post-acute full-time sick leave after a positive test compared with a negative test (RD 3.3%).”

5. In the results section (lines 71-78), could you clarify if the tested positive and negative cohorts are similar or different in terms of patient characteristics? You present their characteristics in Table

¹ <https://www.dst.dk/en/Statistik/> (accessed Aug 04, 2023)

S1 and report them in the results, but it is not explicitly mentioned whether both cohorts are similar or different, and for which characteristics.

Response: The similarity of the tested positive and tested negative cohorts has been clarified in the results subsection on cohort characteristics (lines 82-89): “Overall, the test-positive and test-negative cohorts are similar across the participant characteristics (see Model performance). The test-positive cohort has a higher proportion of males (38.6% vs. 33.6%) and is on average 2.2 years younger (43.6 vs. 45.8 years). In terms of pre-existing health conditions, the test-positive cohort has a smaller proportion of chronic fatigue syndrome (1.3% vs. 1.7%) and chronic obstructive pulmonary disease (1.1% vs. 1.5%), a larger proportion of chronic asthma (7.6% vs. 6.8%), and similar proportions of fibromyalgia (0.9% vs. 0.9%) and post-traumatic stress disorder (2.0% vs. 1.9%). All participant characteristics by exposure group can be found in Table S1.”

6. Regarding Figure S2, could you provide two histograms overlaying each other, one for the tested positive group and one for the tested negative group? It would also be helpful to show the overlap before and after IPTW weighting.

Response: The density plot (now figure S4) has been updated to show the distribution of propensity scores in each exposure group, both before and after applying inverse propensity weights. To the last part of the comment, we point out that the overlap is not influenced by weighting methods as they scale the influence of existing data and cannot change the support. Instead, the IPW method helps with covariate balance, as demonstrated in figure S5-S8.

7. In the methods and results sections, could you describe the distribution of sick leaves? For instance, how many individuals (n and %) took 4, 5, 6, ... weeks of leave? Providing this descriptive distribution would be useful for the readers.

Response: A table (Table 2) has been added to the manuscript with how many individuals responded to the different options given in the questionnaire. The distribution is described in a new results subsection *distribution of sick leave* (lines 91-99): “From the study cohort, 7955 (9.0%) reported taking some amount of full-time sick leave more than 4 weeks after their test date. Of these, 2412 (30.3%) took substantial sick leave of more than 4 weeks. Shorter durations of sick leaves were reported more often than long durations. 346 people reported being on full-time sick leave for more than 9 months or since their test date. Females reported full-time sick leave of any duration more often than males (9.7% of females vs. 7.6% of males). Similarly, individuals 50 years

or older at the time of their test reported taking more full-time sick leave of any duration than individuals below 50 years at the time of their test (9.7% of ≥ 50 years vs 8.3% of < 50 years). The distribution of the duration of self-reported full-time sick leave taken more than 4 weeks after the test date can be found in Table 2.”

8. In the discussion section, it would be valuable to comment on the generalizability of these findings to other countries.

Response: The following discussion on generalizability has been added to the discussion section (lines 210-215): “The transportability of our findings to other countries’ populations may be most relevant when considering how having a combination of risk factors (specifically older age, high BMI, depression, and female sex) can collectively impact one’s health, even after the acute COVID-19 infection. While we were able to examine the burden with respect to post-acute sick leave, other countries may be able characterize this burden in other ways, such as by examining differences in employment termination or disability payments.”

9. In the results section (minor point), please number the tables and figures consecutively. For example, mention Figure S5 and S6 after Figure S2, S3, and S4.

Response: The numbering of tables and figures has been updated so they appear consecutively in the manuscript.

Referee #2:

In the paper "Identifying heterogeneity in the effect of COVID-19 on long-term sick leave using causal machine learning", the authors develop a model to predict the conditional risk difference for an individual to take a long-term sick leave after a SARS-CoV-2 infection compared with not being infected using methods of causal ML and study the heterogeneity of this risk. For that purpose, they rely on the data of a hybrid survey and a register-based cohort study. The paper may provide some tools to assess certain aspects of the socio-economic cost of long covid.

The paper is well-written and well-structured. I have several important methodological comments and remarks:

1/The inclusion/exclusion criteria of the study used are not reported, which should be corrected.

Response: We thank reviewer 2 for their careful consideration of our manuscript and appreciate the opportunity to reflect on- and respond to their comments.

The methods subsection on data sources and study population has been adjusted to clearly indicate the inclusion/exclusion criteria used to construct the study cohort (lines 291-302): “The present study includes participants who responded to a retrospective questionnaire sent out 9 months after their test date. An individual was eligible for invitation to the retrospective questionnaire if they were alive and living in Denmark 9 months after the test date, had a first positive RT-PCR test or a negative RT-PCR test taken between November 4, 2020 and February 1, 2021, didn’t have a positive test result within 9 months of the test date, and was registered with the national mail system, e-Boks, used by 90% of Danish residents aged ≥ 15 years. Invitations were sent out to all eligible individuals with a positive test result, while test-negative controls were randomly selected using incidence density sampling on the test date with a ratio of 2:3 between test-positive and -negative persons. After receiving an invitation, participants were excluded if they failed to complete the questionnaire, indicated they believed they had been previously infected with SARS-CoV-2 due to receiving a seropositive result for SARS-CoV-2, or were >65 years-old at the time of the test.” Additionally, a flowchart (Figure 1) has been added detailing how many are excluded at the different stages.

2/Partly due to the abovementioned issue, it is impossible to determine if all the subjects considered in the study are eligible for the outcome. For instance, it is unclear if unemployed persons or students are eligible for long-term sick leave and how they have been accounted for in the study. This should be discussed and it may be necessary to conduct a sensitivity analysis.

Response: As a rule, all adult in Denmark are eligible for sick leave, including students and unemployed. Students can get leave from their studies, allowing them to take exams at a later date. People who are unemployed are entitled to special social services if they are sick. As described in the methods subsection *outcome ascertainment*, the outcome is self-reported and not based on receiving special social services or on medical records. As such, no special consideration has been given to participants who were unemployed or studying in the follow-up period. A section has been added to the discussion addressing the choice of outcome and potential problems with eligibility (lines 221-230): “There are some limitations with our choice of outcome. One problem is that sick leave only measures those with a connection to the labour market. We note that this includes students and the unemployed. In Denmark, persons who are unemployed can only receive

unemployment benefits if they are actively looking for a job. If a person is unable to comply with their obligations due to illness or injury, they can apply for sick leave.² Similarly, students can apply for sick leave if illness or injury prevents them from attending their exams. Due to the design of the EFTER-COVID questionnaire we cannot rule out the presence of persons ineligible for the outcome in the study cohort, such as persons on early retirement. We presume these persons will not report taking sick leave, since they have no incentive for false reporting, although some could have misinterpreted the question. Overall, we expect the vast majority of the study population to be eligible for the outcome.”

3/There may be censoring regarding the length of the sick leave. For example, how has someone who has been on sick leave for three weeks (and is still on sick leave) at nine-month after its positive SARS-CoV-2 positive test been accounted for in the study?

Response: For this study, we look at the period from 1-9 months after the test date and define substantial sick leave as more than 4 weeks in this period. It is true that some participants may have started their sick leave towards the end of the 9-month period, and some may still be on sick leave when they fill out the survey. Such cases are not specifically accounted for. Because of the design of the EFTER-COVID survey, we only have access to the amount of sick leave taken after 4 weeks from the test and up to the survey response date, not when it was taken, so we do not have a way of ascertaining if censoring has occurred. However, PCC sick leave literature doesn't suggest symptoms which could be responsible for sick leave would have a delayed onset of 8 or more months. People with worse acute symptoms generally have higher risk of PCC,³ suggesting PCC related sick leave mostly starts early. Therefore, we find it less likely that a significant amount of PCC related substantial sick leave is censored due to the time between the test date and answering the questionnaire.

4/The causal groups (infected/not infected) are not as well defined as they may seem. The RT-PCR tests have a particular specificity and sensitivity, resulting in false positive or false negative individuals. It would be interesting to conduct a sensitivity analysis to evaluate if this influences the results or, at the very least, discuss it.

Response: We evaluated the impact of false RT-PCR test results by rerunning our analysis 20 times

² <https://www.borger.dk/arbejde-dagpenge-ferie/Dagpenge-kontanthjaelp-og-sygedagpenge/sygedagpenge/sygedagpenge-hvis-du-er-ledig> (accessed Aug 29, 2023)

³ Davis HE, McCorkell L, Vogel JM, et al. Long COVID: major findings, mechanisms and recommendations. *Nat. Rev. Microbiol.* 2023;21(3):133–146.

using a modified exposure. Binny et.al.⁴ found RT-PCR sensitivity stays above 88% in the first two weeks post-infection, with a peak of 92.7%. Skittrall et.al.⁵ report high specificity above 99%. To evaluate the impact of false test results, we assumed a fixed sensitivity of 90% and a fixed specificity of 99%. Using these values, we solved for the number of false negatives (FN = 4112) and -positives (FP = 477) in the study population. These labels were assigned at random and the modified exposure had TP + FN in the exposed group, and TN + FP in the unexposed group.

The analysis showed a slight reduction in RD estimates across the board, e.g. from 3.3 percent to 2.9 percent in the full population. The five most important risk factors according to the variable importance measure stayed the same, with education ranked higher than sex and depression in some iterations. RDs across age, high BMI, and depression showed an increase with each covariate almost identical to the main analysis. Again, the RDs tend to be slightly lower compared to the main analysis.

In conclusion, the sensitivity analysis did not provide reason to be concerned about results on heterogeneity in the effect of COVID-19 on post-acute sick leave, but it did show a small overestimation of the increased risk of full-time sick leave after COVID-19 infection. We have included this sensitivity analysis in the supplementary materials.

5/It seems that the data do not include information regarding whether individuals had already been on long-term sick leave prior to their confirmed SARS-CoV-2 infection. Such a covariable could be an excellent predictor of post-SARS-CoV-2 long-term sick leave. This situation should at least be thoroughly discussed in the paper if it is impossible to adjust/conduct a sensitivity analysis on this covariable.

Response: We are unfortunately unable to adjust for long-term sick leave prior to their test date, since participants in the EFTER-COVID survey were not asked about this. A discussion of this situation has been added to the manuscript (lines 231-241). Only in the situation where prior full-time sick-leave is associated with the test result, will this introduce bias in our results. In this case participants on prior full-time sick leave may to some extent be captured by the Charlson comorbidity index.

⁴ Binny RN, Priest P, French NP, *et al.* Sensitivity of Reverse Transcription Polymerase Chain Reaction Tests for Severe Acute Respiratory Syndrome Coronavirus 2 Through Time. *J Infect Dis* 2023; **227**: 9.

⁵ Skittrall JP, Wilson M, Smielewska AA, *et al.* Specificity and positive predictive value of SARS-CoV-2 nucleic acid amplification testing in a low-prevalence setting. *Clin Microbiol Infect* 2021; **27**: 469.e9.

Furthermore, the most likely association would be that prior sick leave was associated with a lower probability of testing positive due to decreased social activities. Since persons already on sick-leave are more likely to report substantial sick leave after infection, these assumptions imply that our reported risk difference estimates are conservative compared to what we would get if we could adjust for prior sick-leave, since people with high risk of long-term sick leave end up overrepresented in the test-negative group.

6/The authors use a causal forest to estimate heterogeneity, applying standard metrics to ensure it is well-tuned. They also indicate that the automatic method for choosing the hyperparameters is unstable. Therefore, it would be interesting for them to confirm their findings using another class of model. This is especially important since importance techniques (in this case, the number of splits along a given covariate which is specific to causal forests) are highly dependent on the model's quality. Inadequate models can lead to incorrectly declaring important a variable that is not important and reciprocally.

Response: In the manuscript we use a double robust asymptotically efficient summary of the CATEs in the AIPW estimator to validate our findings on important subgroups. The AIPW estimator is well-established and we compared our results from adjusting the CATE estimates obtained from the causal forest model to the results from an analysis performed on the same data using parametric g-computation with outcomes predicted using logistic regression. Results between the two approaches were similar. Thus, even if our importance metric would have failed to correctly declare the most important variables in terms of effect heterogeneity, we argue that important heterogeneity still exists in the subgroups we report on.

However, we agree that it would be valuable to confirm our findings using a different modelling approach, particularly about treatment effect heterogeneity and identification of important subgroups. Results from such an analysis would be valuable to guide future use of importance techniques for subgroups identification. Unfortunately, we know of no alternative accessible method to replace our causal forest analysis with the goal of identifying subgroups with high causal risk differences and providing patient-centered estimates of excess risk that take multimorbidity into account. In any case, due to the novelty of the causal forest method, a lot of work has gone into exploring the use of the method to identify novel sub-groups at high excess risk, and we feel another analysis of equivalent scope would be deserving of its own paper. A revisit of the data with

an alternative method would be an interesting avenue of future research, as would attempts to replicate our results on new data using the methods presented in our paper.

For the present paper, we have added a sensitivity analysis to the supplementary materials exploring the stability of the causal forest model to hyperparameter tuning by rerunning the analysis over a grid of different parameter values. We did not tune the parameters controlling split imbalance, because excluding splits on low prevalence health conditions causes their variable importance to be 0, and CRDs will be constant along the low prevalence health conditions, irrespective of existing heterogeneity. This issue is a major source of the substantial variation in results cited in the original manuscript. When varying over the remaining tuning parameters, the results showed significant variability in the coefficient of the differential forest prediction (average 0.83, min 0.49, max 1.13) from the best linear fit calibration method. This coefficient indicates how well-calibrated the heterogeneity estimates from the causal forest are. Despite this variability, ranking of the five most important risk factors was consistent with the main analysis and RDs varied minimally across choices of hyperparameters. Overall, there was no evidence that the estimated RDs were sensitive to tuning of hyperparameters in the causal forest algorithm when excluding parameters controlling the imbalance between child leaves in a split. The limitations discussion has been updated to reflect the results from the sensitivity analysis (lines 262-277).

7/ There is a closing parenthesis missing on the left panel of Figure 2 around percentage points

Response: The closing parenthesis has been added.

8/Same comments for Figure S5.

Response: Please see answer to comment 7.

REVIEWERS' COMMENTS:

Reviewer #1 (Remarks to the Author):

Thank you for addressing all the comments.

Reviewer #2 (Remarks to the Author):

The authors have answered all my questions thoroughly and precisely. I have no additional comments and recommend the paper for publication.